analytical chemistry/materials science/physical chemistry

polyethylene terephthalate, certified reference material, oxygen gas transmission, manometric method, coulometric method, uncertainty evaluation

**Author for correspondence:**
Hui-Min Sun
e-mail: sunhm@126.com

This article has been edited by the Royal Society of Chemistry, including the commissioning, peer review process and editorial aspects up to the point of acceptance.

# A new standard reference film for oxygen gas transmission measurements

Lan-Gui Xie, Xia Zhao, Si-Hong Dou, Long Tang and Hui-Min Sun

The Institute for Packaging Materials and Pharmaceutical Excipients Control, National Institutes for food and drug Control, 2 Tiantan Xi Li, Beijing 100050, People's Republic of China

(iD) L-GX, 0000-0003-4567-2268; H-MS, 0000-0001-5010-3698

A novel reference film was characterized to improve the oxygen gas transmission measurement accuracy of plastic materials for pharmaceutical packaging. The material processing, homogeneity, stability, jointly determined value and uncertainty evaluation were discussed. The film is the first reference film characterized by multiple laboratories using both manometric and coulometric methods. The oxygen transmission rate of the reference film was 20.53 with the expanded uncertainty of 1.36. The newly characterized reference film can be used in the calibration and self-calibration of oxygen transmission measurement equipment and analytical method verification to improve the measurement accuracy and achieve traceable data.

## 1. Introduction

Oxygen gas transmission rate is one of the most important parameters for evaluating the barrier properties of food and drug packaging materials [1]. It is a key factor involved in the quality control of packaging materials [2,3]. Oxygen gas transmission rate is usually measured by the manometric method [4] described in ISO 15105-1:2007 and the coulometric method [5,6] published in the ISO 15105-2:2003. However, the data published by the American Society for Testing and Materials [7] and those we collected in China in 2014 (table 1) suggest that the oxygen transmission rates obtained by different laboratories using different methods exhibit significantly statistical dispersion and abnormal distributions (figure 1). The significant inter-laboratory variations are unconducive to the data comparison, product quality analysis and quality control. Therefore, reducing the inter-laboratory variation in oxygen gas transmission measurements is highly urgent to produce reliable and comparable data.

**Figure 1.** Statistical dispersion of oxygen transmission rates shown in table 1. The horizontal and vertical coordinates represent the oxygen transmission rates reported by 22 laboratories and the referred laboratory number of the corresponding oxygen transmission rate, respectively. The statistic histogram shows that measurement data from 22 laboratories are not a normal distribution.

**Table 1.** Oxygen transmission rates of the same plastic film obtained with different instruments and different methods from 22 Chinese laboratories in 2014.

| laboratory number | oxygen gas transmission rates (ml m$^{-2}$ d$^{-1}$) |
| --- | --- |
| L08 | 79.4 |
| L32 | 85.5 |
| L11 | 87.2 |
| L18 | 87.2 |
| L14 | 87.4 |
| L12 | 90.4 |
| L01 | 102.7 |
| L31 | 104.3 |
| L24 | 108.0 |
| L07 | 108.7 |
| L05 | 114.7 |
| L02 | 117.1 |
| L26 | 118.7 |
| L04 | 120.1 |
| L03 | 120.9 |
| L06 | 121.9 |
| L25 | 122.0 |
| L09 | 122.3 |
| L16 | 122.4 |
| L27 | 122.7 |
| L15 | 123.0 |
| L30 | 123.8 |
| average value | 108.0 |
| RSD | 14.4%[a] |

[a]RSD 14.4% suggests the significant dispersion of the oxygen transmission rates obtained by different laboratories using different methods. In general, the RSD of oxygen transmission rates for a homogeneous plastic film should not be higher than 10% and the data should be normally distributed.

Such significant inter-laboratory variations are due to different self-calibration films for their instruments. Most manufacturers of oxygen gas transmission measurement instruments provide self-calibration films either for manometric method or for coulometric method, based on the measurement principle. These calibration films are given different reference values because of the different testing principles and conditions. For example, the manometric method is conducted with one atmosphere pressure difference between the two sides of the film and the coulometric method requires the same pressure on the two sides of the film.

Despite the different testing conditions, the behaviour of gas transport through the polymeric membrane is dominated by the solution–diffusion model [8,9] and the driving forces of the oxygen gas transmission for both methods are the same, e.g. chemical potential gradient [10]. The seemingly different testing principles and conditions turn out to be identical. Limited by the manometric current instrument design, the chemical potential that should be maintained constant gradually becomes smaller during the manometric test, while that during the coulometric test remains constant. Therefore, the results of the manometric method are usually lower than those of the coulometric method, consistent with the variations between the reported data [11]. To exclude the effect of chemical potential deviation, the results of manometric tests should be calibrated with a reference film.

Both manometric and coulometric oxygen gas transmission measurements under certain conditions are simulation tests correlated with the practical oxygen barrier properties of the polymeric film. In the practical applications, different calibration films with different reference oxygen gas transmission values are inconsistent, which have long confused the producers and users of packaging materials. Technically, the oxygen gas transmission rates measured by those two methods should be consistent. So far, no appropriate universal reference film is available for all measurement methods and instruments. Developing a universal standard reference film for calibration and self-calibration of most instruments is an urgent task.

In the present work, a novel reference film for the oxygen gas transmission measurements was characterized based on ISO Guides 34 and 35 [12–20]. The film developments, including material processing, homogeneity, stability, joint determination and uncertainty evaluation, were discussed. The value of the reference film was determined by multiple laboratories using both manometric and coulometric methods for the first time. It was found that the reference film was accurate with ideal uncertainty and it was suitable for the calibration and self-calibration of most instruments on the market. The reference film would significantly improve the accuracy of these instruments to achieve comparable and traceable data.

# 2. Material and methods

## 2.1. Instruments

An OX-Tran 2–21 oxygen transmission rate tester (Mocon, USA), Brugger gas transmission tester (Germany), N530 gas transmission analyzer (Guangzhou Biaoji Packaging Equipment Co. Ltd., China), Vac-V2gas permeability tester (Jinan Labthink, China) and Filmetrics F70 thin film analyzer were used in the present work.

## 2.2. Materials

The polyester film with a size of 1000 ml × 1 m W and a thickness of 80 μm was purchased from Toray Industries (Japan). The reference film for the F70 thin film analyzer with a thickness of 222.28 μm was provided by Filmetrics. The reference films with oxygen transmission rates of 0.2452 ml pkg d$^{-1}$, 0.0531 ml pkg d$^{-1}$ and 0.01042 ml pkg d$^{-1}$ with reference to CRM1470 film were purchased from Mocon.

## 2.3. Preparation of the candidate CCRM195017

To select an ideal reference film material, a number of quality indexes including the amount, homogeneity and stability of the oxygen transmittance, the homogeneity of the thickness were set to screen the most popular plastic films on the market. The candidate film was chosen with a homogeneous and stable oxygen transmittance of approximately 20 ml m$^{-2} \cdot$ day and a homogeneous thickness and named as CCRM195017. Due to the different test areas of the oxygen transmission testers produced by different instrument manufacturers, the polyester film was cut into uniform square pieces with a size of 15 cm × 15 cm. The edges of 5 cm of the polyester film were discarded to exclude defects.

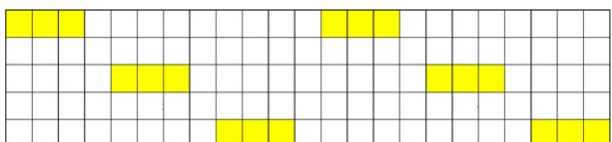

**Figure 2** Representative sample selection on the film of six units. The dark squares are the sampling units and each sample unit includes three square pieces.

## 2.4. Determination of homogeneity and stability of the candidate CCRM195017

The homogeneities of thickness and oxygen transmission of the polyester film were evaluated by between-group and within-group tests. According to the ASTM D1898, a sampling protocol was self-designed to represent the maximum statistical variation of the whole film (figure 2) [21]. The between-group homogeneity was determined with 15 selected sampling units. The within-group homogeneity was determined with three films in the same sampling unit. The homogeneity of the CRM film was evaluated by one-way analysis of variance (ANOVA).

The short-term and long-term stabilities of the polyester film were evaluated. Thirty samples were equally divided into two groups, which were stored at 60°C and −20°C. Their oxygen transmission rates were measured at days 0, 3, 7, 11 and 14 to evaluate the short-term stability of the film. Thirty-two randomly selected samples were stored at 25°C and measured for their oxygen transmission rates at months 0, 3, 6 and 18 to examine the long-term stability of the film. The experimental data were analysed by linear regression to evaluate the stability of the candidate reference film using the slope changes of the linear curve.

## 2.5. Characterization of the candidate CCRM195017

The reference film was tested by four CNAS (China National Accreditation Service for Conformity Assessment) laboratories using both manometric and coulometric methods. The film samples were kept in a desiccator at $23 \pm 2$°C for 48 h before use. All instruments were conducted to be calibrated with the reference films from Mocon before the tests. The test was conducted at 23°C.

### 2.5.1. Manometric method measurements

Before each test, the seal of the sample holder was coated with a thin layer of vacuum grease. The film sample was placed on the sample holder and gently pressured to ensure a good contact with the vacuum grease. The test chamber was then closed to initiate the test. The high-pressure and low-pressure chambers were then vacuumed until the pressure of the low-pressure chamber reached lower than 20 Pa. High-purity oxygen was then slowly introduced into the high-pressure chamber until the pressure reached $1.01 \times 10^5$ Pa. The oxygen in the high-pressure chamber then gradually permeated the film into the low-pressure chamber, which was monitored by a pressure sensor in the low-pressure chamber. The test was stopped as the pressure difference remained constant at the same time intervals. The data recorded by the pressure sensor were then used to calculate the oxygen volume permeated through per unit area of the film. The oxygen transmission rate was defined as the oxygen volume permeated through unit area of the film per unit time.

### 2.5.2. Coulometric method measurements

Before each test, the seal of the sample holder was coated with a thin layer of vacuum grease. The film sample was placed on the sample holder and gently pressured to ensure a good contact with the vacuum grease. The test chamber was then closed to initiate the test. One side of the film was exposed to oxygen and the other side was exposed to nitrogen. The oxygen permeated through the film caused a chemical reaction in the coulometer, inducing a voltage that was recorded and used to calculate the oxygen transmission rate. The test was stopped as the voltage recorded by the coulometer became constant at the same time interval. The oxygen transmission rate was defined as the oxygen volume permeated through per unit area of the film per unit time.

**Table 2.** Thickness homogeneity of film CCRM195017.

| group no. | thickness (μm) | | |
| --- | --- | --- | --- |
| | test no. 1 | test no. 2 | test no. 3 |
| 1 | 82.24 | 82.74 | 81.56 |
| 2 | 81.77 | 82.35 | 82.77 |
| 3 | 83.14 | 82.68 | 82.47 |
| 4 | 82.89 | 82.23 | 82.20 |
| 5 | 82.36 | 82.44 | 81.22 |
| 6 | 82.57 | 81.92 | 82.09 |
| 7 | 81.67 | 82.70 | 82.61 |
| 8 | 81.79 | 82.31 | 81.25 |
| 9 | 82.57 | 82.49 | 82.70 |
| 10 | 81.34 | 82.93 | 81.94 |
| 11 | 81.94 | 81.55 | 81.87 |
| 12 | 82.34 | 82.76 | 82.25 |
| 13 | 81.56 | 82.68 | 82.63 |
| 14 | 82.40 | 81.41 | 82.22 |
| 15 | 82.55 | 83.02 | 82.83 |
| average value | | 82.27 | |
| RSD (%) | | 0.61 | |

## 3. Results

### 3.1. Homogeneity of the candidate CCRM195017

The homogeneities of both thickness (table 2) and oxygen gas transmission (table 3) of the candidate CCRM195017 were evaluated by $F$-test. The data were first examined by Grubbs test and no outlier was found. The data exhibited a normal distribution and thus were analysed by ANOVA. The mean squares between groups and within group of the thickness were 0.290 and 0.238, respectively. The $F$-value of the film thickness was 1.22, lower than the critical $F$-values (2.04), indicating that the thickness of the candidate CCRM195017 is homogeneous. The mean of squares between groups and within group of the oxygen gas transmission rate were 0.337 and 0.184, respectively, and the $F$-value was 1.83, lower than the critical $F$-values (2.04), suggesting that the oxygen gas transmission of the candidate CCRM195017 is homogeneous.

### 3.2. Stability of the candidate CCRM195017

The short-term and long-term stability were determined by linear regression analysis to evaluate the stability of the film during transportation and storage, respectively. No significant difference was found in the oxygen gas transmission rates of the film after storage at 60°C (table 4) and −20°C (table 5), for two weeks or at room temperature for 18 months. The statistical analysis of the oxygen gas transmission rates obtained during 18 months' storage at room temperature (table 6) showed a linear slope of −0.0029, not significantly different from the expected value (0). Therefore, the polyester film is stable at room temperature for at least 18 months.

### 3.3. Value assignment

Eight datasets obtained by the manometric and coulometric methods were examined by the Grubbs and Dixon tests and no outlier was found (table 7). Therefore, all data were included for normal distribution testing. The results indicate the data exhibited a normal distribution. The average values of four datasets

**Table 3.** Oxygen gas transmission homogeneity of film CCRM195017.

| group no. | oxygen gas transmission rates (ml m$^{-2}$ (0.1 MPa d)$^{-1}$) | | |
| --- | --- | --- | --- |
| | test no. 1 | test no. 2 | test no. 3 |
| 1 | 18.719 | 19.576 | 20.078 |
| 2 | 20.793 | 19.821 | 20.019 |
| 3 | 20.345 | 20.017 | 19.021 |
| 4 | 20.363 | 20.218 | 19.023 |
| 5 | 20.488 | 20.526 | 21.153 |
| 6 | 19.894 | 19.875 | 20.309 |
| 7 | 19.812 | 18.744 | 19.579 |
| 8 | 20.675 | 19.857 | 20.705 |
| 9 | 19.760 | 19.773 | 19.992 |
| 10 | 20.686 | 19.966 | 19.579 |
| 11 | 19.981 | 20.897 | 20.007 |
| 12 | 19.962 | 20.609 | 19.658 |
| 13 | 20.495 | 19.335 | 19.586 |
| 14 | 21.014 | 20.710 | 20.609 |
| 15 | 19.685 | 19.956 | 20.367 |
| average value | 20.050 | | |
| RSD (%) | 2.85 | | |

**Table 4.** Short-term stability of film CCRM195017 at 60°C.

| storage time (day) | oxygen gas transmission rates (ml m$^{-2}$ (0.1 MPa d)$^{-1}$) | | | |
| --- | --- | --- | --- | --- |
| | test no. 1 | test no. 2 | test no. 3 | average value |
| 0 | 20.251 | 21.166 | 20.582 | 20.666 |
| 3 | 19.854 | 19.794 | 19.579 | 19.742 |
| 7 | 20.095 | 20.245 | 19.592 | 19.977 |
| 11 | 19.806 | 19.811 | 19.793 | 19.804 |
| 14 | 20.078 | 20.295 | 19.972 | 20.115 |

**Table 5.** Short-term stability of film CCRM195017 at −20°C.

| storage time (day) | oxygen gas transmission rates (ml m$^{-2}$ (0.1 MPa d)$^{-1}$) | | | |
| --- | --- | --- | --- | --- |
| | test no. 1 | test no. 1 | test no. 1 | average value |
| 0 | 18.964 | 19.957 | 19.517 | 19.480 |
| 3 | 20.059 | 20.467 | 19.959 | 20.162 |
| 7 | 20.907 | 20.581 | 20.301 | 20.596 |
| 11 | 20.039 | 20.089 | 20.187 | 20.105 |
| 14 | 19.589 | 19.920 | 20.360 | 19.957 |

obtained by the manometric method and those obtained by the coulometric method were examined by Dixon test and no outlier was found. Therefore, all data should be retained.

The average of the oxygen gas transmission rates measured by the four laboratories using the manometric method is 20.61 and that measured by the four laboratories using the coulometric

**Table 6.** Long-term stability of film CCRM195017 at 25°C.

| | oxygen gas transmission rates (ml m$^{-2}$ (0.1 MPa d)$^{-1}$) | | | |
|---|---|---|---|---|
| test no. | 0 month | the 3rd month | the 6th month | the 18th month |
| 1 | 20.74 | 20.83 | 21.34 | 20.32 |
| 2 | 20.79 | 20.32 | 21.18 | 20.64 |
| 3 | 20.33 | 19.41 | 20.78 | 20.42 |
| 4 | 20.83 | 19.77 | 20.86 | 20.76 |
| 5 | 20.77 | 20.79 | 21.00 | 21.00 |
| 6 | 20.54 | 20.76 | 21.26 | 20.65 |
| 7 | 20.89 | 20.54 | / | / |
| 8 | 20.84 | 20.84 | / | / |
| 9 | 20.75 | 20.75 | / | / |
| 10 | 20.89 | 20.89 | / | / |
| average | 20.74 | 20.49 | 21.07 | 20.63 |

**Table 7.** Oxygen gas transmission rates of film CCRM195017 measured by 4 CNAS laboratories using the manometric method or the coulometric method.

| | oxygen gas transmission rates measured by the manometric method (ml m$^{-2}$ (0.1 MPa d)$^{-1}$) | | | | oxygen gas transmission rates measured by the coulometric method (ml m$^{-2}$ d$^{-1}$) | | | |
|---|---|---|---|---|---|---|---|---|
| test no. | M1 lab | M2 lab | M3 lab | M4 lab | M1 lab | M2 lab | M3 lab | M4 lab |
| 1 | 21.3 | 20.2 | 20.66 | 19.738 | 20.92 | 20.02 | 20.17 | 20.743 |
| 2 | 21.7 | 20.6 | 19.90 | 19.676 | 20.01 | 20.16 | 20.20 | 20.790 |
| 3 | 22.3 | 20.4 | 20.35 | 20.241 | 20.40 | 20.64 | 20.60 | 20.326 |
| 4 | 20.4 | 19.8 | 21.16 | 20.166 | 20.48 | 20.76 | 20.40 | 20.834 |
| 5 | 21.2 | 20.5 | 20.27 | 19.654 | 21.02 | 20.32 | 20.35 | 20.765 |
| 6 | 22.1 | 20.4 | 20.62 | 19.803 | 20.68 | 20.84 | 20.26 | 20.540 |
| 7 | 22.0 | 19.5 | 19.63 | 20.993 | 20.16 | 20.88 | 20.22 | 20.892 |
| 8 | 21.0 | 20.4 | 21.04 | 19.911 | 20.43 | 21.05 | 20.15 | 20.844 |
| 9 | 22.3 | 20.9 | 20.51 | 20.922 | 19.89 | 20.87 | 20.22 | 20.754 |
| 10 | 22.2 | 20.7 | 19.86 | 19.538 | 19.73 | 20.97 | 20.26 | 20.892 |
| mean | 21.65 | 20.34 | 20.40 | 20.06 | 20.37 | 20.65 | 20.28 | 20.74 |
| s.d. | 0.65 | 0.42 | 0.50 | 0.52 | 0.43 | 0.36 | 0.14 | 0.18 |
| assigned value | 20.53 | | | | | | | |

method is 20.51. The oxygen gas transmission rates determined by the manometric and coulometric methods were examined by t test and no significant difference between them was found. The data obtained by different methods are unequally precise measurements. Their average should be calculated as the weight arithmetic mean. Therefore, the oxygen gas transmission rate of the film was calculated as follows:

$$\overline{\overline{x}} = \frac{\sum_{i=1}^{m} W_i \overline{x_i}}{\sum_{i=1}^{m} W_i} = 20.53 \tag{3.1}$$

and

$$W_i = \frac{1}{(S_i/\sqrt{n_i})^2} , \tag{3.2}$$

where $\overline{\overline{x}}$ is the weighted arithmetic mean, $W_i$ is the weight of each method, $\overline{x}_i$ is the mean of each method, $S_i$ is the variance of each method and $n_i$ is the number of data obtained by each method.

## 3.4. Evaluation of uncertainties

The uncertainty of the oxygen gas transmission of CCRM195017 is derived from three sources: homogeneity, stability and the joint determination based on the ISO/IEC Guide 98-3:2008 [22,23].

### 3.4.1. Uncertainty of homogeneity

The uncertainty of homogeneity was calculated as follows.

$$u_H = \sqrt{\frac{MS_{among} - MS_{within}}{n}} = \sqrt{\frac{0.337 - 0.184}{3}} = 0.23. \tag{3.3}$$

### 3.4.2. Uncertainty of stability

The variation of the linear slop ($S_b$) of the stability test at room temperature for 18 months was 0.02. The uncertainty of the 18-month storage was then calculated to be 0.36 as shown below.

$$u_S = S_b \times T = 0.02 \times 18 = 0.36. \tag{3.4}$$

### 3.4.3. Uncertainty of joint determination

The uncertainty of joint determination of the oxygen gas transmission of the CCRM195017 by multiple laboratories includes two parts. The first part is the uncertainty of the statistical calculation as shown below.

$$u_{\overline{\overline{x}}} = \pm t_{\alpha(m-1)} \sqrt{\frac{\sum_{i=1}^{m} W_i v_i^2}{(m-1)\sum_{i=1}^{m} W_i}} = \pm 0.47 . \tag{3.5}$$

The second part is the uncertainty caused by the variations of the factors affecting the measurement and determined by non-statistical analysis.

#### 3.4.3.1. Uncertainty of manometric method ($uc(P_{MM})$)

The oxygen gas transmission rate ($Q_g$) can be calculated using the following equation:

$$Q_g = \frac{\Delta P}{\Delta t} \times \frac{V}{S} \times \frac{T_0}{P_0 T} \times \frac{24}{P_1 - P_2}, \tag{3.6}$$

where $\Delta P$ is the pressure change in the low-pressure chamber, $\Delta t$ is the time interval, $V$ is the volume of low-pressure chamber, $S$ is the gas permeation area of the film, $T$ is the temperature of the test chambers, and $P_1 - P_2$ is the pressure difference between the two sides of the film. Therefore, the uncertainty caused by the variations of the six components (table 8) was calculated as follows, which contributed to the second part of the uncertainty of the joint determination.

$$\frac{u_c(P_{MM})}{P_{MM}} = \sqrt{\left(\frac{u_c(\Delta P)}{\Delta P}\right)^2 + \left(\frac{u_c(\Delta t)}{\Delta t}\right)^2 + \left(\frac{u_c(V)}{V}\right)^2 + \left(\frac{u_c(S)}{S}\right)^2 + \left(\frac{u_c(T)}{T}\right)^2 + \left(\frac{u_c(P_1 - P_2)}{P_1 - P_2}\right)^2}$$
$$= 0.0149. \tag{3.7}$$

#### 3.4.3.2. Uncertainty of the coulometric method ($u_c(P_{CM})$)

The variations of the three variables (table 9) affecting the coulometric test were used to calculate the uncertainty of the method using the equation below, which also contributed to the second part of the

**Table 8.** Variations of the components affecting the manometric test.

| component | description | value | standard derivation | relative standard derivation |
|---|---|---|---|---|
| $\Delta P$ | pressure change in the low-pressure chamber | 25.87 Pa | 0.2582 Pa | 0.010 |
| $\Delta t$ | time interval | 30 min | 0 min | 0 |
| $V$ | volume of the bottom chamber | 5.30 cm$^3$ | 0.0053 cm$^3$ | 0.001 |
| $S$ | transmission area | 50.24 cm$^2$ | 0.01 cm$^2$ | 0.0002 |
| $T$ | chamber temperature | 23°C | 0.23°C | 0.011 |
| $P_1 - P_2$ | pressure difference between two sides of the film | 101.02 KPa | 124.275 Pa | 0.001 |

**Table 9.** Variations of the components affecting the coulometric test.

| component | description | value | standard derivation | relative standard derivation |
|---|---|---|---|---|
| $Sc$ | accuracy of the coulometric sensor | / | / | 0.00577 |
| $S$ | transmission area | 50 cm$^2$ | 0 cm$^2$ | 0 |
| $T$ | chamber temperature | 23°C | 0.2898°C | 0.0126 |

uncertainty of the joint determined.

$$\frac{u_c(P_{\mathrm{CM}})}{P_{\mathrm{CM}}} = \sqrt{\left(\frac{u_c(Sc)}{Sc}\right)^2 + \left(\frac{u_c(S)}{S}\right)^2 + \left(\frac{u_c(T)}{T}\right)^2} = 0.014, \tag{3.8}$$

where $Sc$ is the precision of the coulometric sensor, $S$ is the gas permeation area of the film and $T$ is the test temperature.

### 3.4.4. Uncertainty of value determination

The relative standard derivations of the manometric and coulometric methods were then converted into the standard derivation of value determination as shown below:

$$u_c = \bar{x} \times \sqrt{W_i^2\left(\frac{u_c(P_{\mathrm{MM}})}{P_{\mathrm{MM}}}\right)^2 + W_j^2\left(\frac{u_c(P_{\mathrm{CM}})}{P_{\mathrm{CM}}}\right)^2} = 0.25, \tag{3.9}$$

where $\bar{\bar{x}}$ is the weighted arithmetic mean, $W_i$ and $W_j$ are the weights of each method. The uncertainty of the value determination of the candidate CCRM195017 was then calculated as the square root of the sum of the first part and second part of the uncertainty as shown below:

$$u_c = \sqrt{u_{\bar{\bar{x}}}^2 + u_{\mathrm{C}}^2} = 0.53. \tag{3.10}$$

### 3.4.5. Combined uncertainty of the assigned value

The combined uncertainty of the oxygen gas transmission rate of the candidate CCRM195017 was then calculated by combining the uncertainties of homogeneity ($U_H$), stability ($U_S$) and the joint determination ($U_C$) as shown below:

$$u_c = \sqrt{u_H^2 + u_S^2 + u_C^2} = \sqrt{0.23^2 + 0.36^2 + 0.53^2} = 0.68. \tag{3.11}$$

### 3.4.6. Expanded uncertainty of the assigned value

The expanded uncertainty of the oxygen gas transmission rate of the candidate CCRM195017 was calculated by multiplying the combined uncertainty ($U_c$) with the coverage factor ($k$) as shown below:

$$U_c = u_c \times k = 0.68 \times 2 = 1.36. \tag{3.12}$$

Based on these results, the oxygen gas transmission rate of the candidate CCRM195017 was $20.53 \pm 1.36$.

# 4. Discussion

## 4.1. Traceability of CCRM195017

The reference film for the oxygen gas transmission measurements characterized in this paper was traced to the CRM1470 film through the reference films purchased from Mocon. The US Bureau of Standards (National Bureau of Standards, NBS) CRM1470 (certificated reference material 1470) standard film [24] was the most widely used reference film for oxygen gas transmission measurements in the early days, yet had been discontinued. The value of the CRM1470 standard film was identified using the manometric method. The results of the manometric method were affected by many factors, so it was sometimes difficult to keep the continuity of value delivery. Later, when the coulometric method instrument for measuring oxygen permeability was developed, the value of the CRM1470 standard film was converted into a current signal. The precision and stability of the coulometric method were improved, which was beneficial to the transfer of the value. So currently the calibration film used by the instrument of the coulometric method was traceable to the CRM1470 film, while the calibration film used by the manometric method was not.

## 4.2. Reasons for multi-method calibration of reference film

Presently, there was no reference film certified by both manometric and coulometric methods nor was a universal reference film applied for calibration of most instruments. In this paper, both manometric and coulometric methods were used to certify the oxygen transmission reference film, which took into account the different precision or uncertainty of different methods. Otherwise, the uncertainty of the reference film prepared by one method was not applicable to the other one. Thus, the reference film prepared in this paper was adapted for the calibration and self-calibration of most instruments on the market.

## 4.3. Sampling protocol for homogeneity test

In order to evaluate the homogeneity of the oxygen permeation of the entire roll film, it was necessary to sample and measure the representative part of the membrane material. The conventional sampling scheme was random sampling and stratified random sampling. However, the membrane material was prepared by a rolling method. The random sampling scheme was not as reasonable as the representative area sampling protocol. Therefore, a sampling scheme with both horizontal and longitudinal representation was designed in this paper (figure 2). Each sampling point effectively reflected the different lateral and longitudinal positions, which in turn effectively reflected the representative of sampling from the whole roll membrane.

## 4.4. Auxiliary evaluation index for homogeneity of reference film

The homogeneity of the oxygen gas transmission of a polyester film is affected by numerous factors, such as the homogeneity of the film, the stability of the instrument and the consistency of the experimental operation. Therefore, the RSD of oxygen transmission rate of a film less than 1% is barely achievable. In the present work, the homogeneity of the oxygen gas transmission rates of the candidate CCRM195017 was evaluated with its thickness. In general, the thickness measurement is highly accurate and the thickness of a film is usually correlated with its oxygen gas transmission rate. The highly homogeneous thickness of a film indicates its highly homogeneous oxygen gas transmission rate. Any abnormal result might be caused by the poor performance of the instrument. The oxygen transmission tester, then, should be well maintained and calibrated.

## 4.5. Value assignment and evaluation of uncertainties

Since the measurement values obtained by different methods are unequally precise measurements, their average should be calculated as the weighted arithmetic mean not arithmetic mean. The uncertainty of the oxygen gas transmission of the reference film derived from the joint determination should also be calculated as the weighted arithmetic mean, not arithmetic mean.

# 5. Conclusion

In the present work, a reference film for oxygen gas transmission measurements was characterized upon processing, homogeneity, stability, joint determination and uncertainty. The oxygen gas transmission of the film was measured by multiple laboratories using both manometric and coulometric methods to be $20.53 \pm 1.36$. The stability analysis indicates that the film is stable at room temperature for at least 18 months.

Manometric and coulometric methods are two commonly used methods for measuring oxygen gas transmission. The oxygen transmission rates of the reference film measured with different calibrated or self-calibrated instruments (referred to CRM 1470 film) using these two methods are consistent, indicating that the film meets the requirement for the oxygen gas transmission measurements by both methods.

This new reference film can be used for the calibration and self-calibration of the oxygen transmission testers and the evaluation of test processes to ensure the accuracy of the measurement results and the comparability of the results obtained by different methods and test equipment. The film has been approved as a reference film for drug packaging material by the Institute of China Food and Drug Control and has been marketed with the reference number 195017.

Data accessibility. Data available from the Dryad Digital Repository at: https://doi.org/10.5061/dryad.qj481q1 [25].
Authors' contributions. L.-G.X. carried out the oxygen gas transmission measurement laboratory work, participated in data analysis, carried out the design of the study and drafted the manuscript; X.Z. participated in the design of the study and coordinated the study. S.-H.D. carried out the oxygen gas transmission measurement laboratory work. L.T. coordinated the study. H.-M.S. conceived of, designed and coordinated the study and helped draft the manuscript. All authors gave final approval for publication.
Competing interests. We declare we have no competing interests.
Funding. The authors would like to thank the fund (2017X2) received from NIFDC which supported their research.
Acknowledgements. The authors thank Jinan Quality Inspection Centre for Pharmaceutical Packages of China Food and Drug Administration, Shanghai Food and Drug Packaging Material Control Centre, State Food and Drug Administration Drugs Packing Material Scientific Research Inspection Centre and Zhejiang Institute for Food and Drug Control for their supports.

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
