## [Reviewer comments · Royal Society Open Science]

Review History

RSOS-180806.R0 (Original submission)

Review form: Reviewer 1

Is the manuscript scientifically sound in its present form?

No

Are the interpretations and conclusions justified by the results?

Yes

Is the language acceptable?

Yes

Is it clear how to access all supporting data?

Not Applicable

Do you have any ethical concerns with this paper?

No

Have you any concerns about statistical analyses in this paper?

No

Recommendation?

Reject

Comments to the Author(s)

The authors developed a new standard reference film for oxygen gas transmission measurement. The data showed the material itself is indeed stable and can be used as reference film, however, the problem the authors think they are going to solve may not exist. The fact that they sent their reference film to 4 CNAS laboratories and got consistent results proved the existing reference materials were working properly.

1. Reference 7 talked about a novel PLA/PHBV Films and its characterization (including measuring helium permeability, but not oxygen). It is a little puzzling that the authors used this to draw a conclusion that there is dispersion in measuring oxygen transmission rates by ASTM.

2. Data source for table 1 is not clear? Did the authors send the same plastic film to various labs for testing or obtain data from these labs for the same type of material? If it is the later, maybe they are measuring slightly different materials.

3. Reference 4 didn't mention anything about results from manometric method are lower than coulometric method. The fact that these two methods used different mechanism is exactly the reason why they used calibration films.

Based on 1-3, the problem the authors try to solve may not exist. In fact, any measurement instruments should ship with consistent and known value reference/calibration material, otherwise it's a manufacture issue, not scientific issue.

4. This statement "The standard reference film should be suitable to the calibration and self-calibration for all instruments" is not correct. And reference 12 is not related to topic.

5. Please check all the references, and make sure they are relevant.

Review form: Reviewer 2

Is the manuscript scientifically sound in its present form?

No

Are the interpretations and conclusions justified by the results?

Yes

Is the language acceptable?

Yes

Is it clear how to access all supporting data?

Yes

Do you have any ethical concerns with this paper?

No

Have you any concerns about statistical analyses in this paper?

No

Recommendation?

Major revision is needed (please make suggestions in comments)

Comments to the Author(s)

In this work, a reference film was studied to improve the oxygen gas transmission measurement accuracy of plastic materials for pharmaceutical packaging. The manometric method and coulometric method was first used to test oxygen gas transmission rate of candidate reference film simultaneously. Besides, the homogeneity, stability, jointly determined value and uncertainty evaluation of the reference film were also discussed. However, some revisions should be made to make the paper more convincing.

1. Check the writing format carefully to ensure its correctness. At the 35 lines on the first page and 23 lines on the second page, what is the “.1)” and “.2)” mean? You might want to write “(Figure 1)” and “(Figure 2)”. At the the 43 lines on the second page, “ $1.01 \times 105\text{Pa}$ ” need to be changed to “ $1.01 \times 105\text{Pa}$ ”. In addition, you had better to indent the first line and align the two ends of article.
2. As for the candidate reference film, how did you prepare your material? The preparation is just cut the film that you bought? If the material is not your original, you can't write “A novel reference film was developed”.
3. Please describe your experiment in detail to make sure that others can repeat your experiment. You could name your prepared reference film to distinguish between other materials.
4. In the figure 1, please write the meaning and unit of horizontal and vertical coordinates.
5. Please show all the experiment data using table or graph, incloud stability experiment and homogeneity experiment.

Decision letter (RSOS-180806.R0)

15-Aug-2018

Dear Dr Xie:

Manuscript ID: RSOS-180806

Title: "A new standard reference film for oxygen gas transmission measurements"

Thank you for submitting the above manuscript to Royal Society Open Science. Your paper was sent to reviewers and their comments are included at the bottom of this letter.

In view of the concerns raised by the reviewers, the manuscript has been rejected in its current form. However, a new manuscript may be submitted which takes into consideration these comments.

Please note that resubmitting your manuscript does not guarantee eventual acceptance, and that your resubmission will be subject to peer review before a decision is made.

Your resubmitted manuscript should be submitted by 12-Feb-2019. If you are unable to submit by this date please contact the Editorial Office.

Yours sincerely,
Dr Laura Smith, MRSC
Publishing Editor, Journals
Royal Society of Chemistry,
Thomas Graham House,
Science Park, Milton Road,
Cambridge, CB4 0WF, UK

Royal Society Open Science - Chemistry Editorial Office

On behalf of the Subject Editor Professor Anthony Stace and the Associate Editor Dr Ya-Wen Wang

REVIEWER(S) REPORTS:

Associate Editor Comments to Author ():

RSC Associate Editor:

Comments to the Author:

(There are no comments.)

RSC Subject Editor:

Comments to the Author:

(There are no comments.)

Reviewers' Comments to Author:

Reviewer: 1

Comments to the Author(s)

The authors developed a new standard reference film for oxygen gas transmission measurement. The data showed the material itself is indeed stable and can be used as reference film, however, the problem the authors think they are going to solve may not exist. The fact that they sent their reference film to 4 CNAS laboratories and got consistent results proved the existing reference materials were working properly.

1. Reference 7 talked about a novel PLA/PHBV Films and its characterization (including measuring helium permeability, but not oxygen). It is a little puzzling that the authors used this to draw a conclusion that there is dispersion in measuring oxygen transmission rates by ASTM.
2. Data source for table 1 is not clear? Did the authors send the same plastic film to various labs for testing or obtain data from these labs for the same type of material? If it is the later, maybe they are measuring slightly different materials.
3. Reference 4 didn't mention anything about results from manometric method are lower than coulometric method. The fact that these two methods used different mechanism is exactly the reason why they used calibration films.

Based on 1-3, the problem the authors try to solve may not exist. In fact, any measurement instruments should ship with consistent and known value reference/calibration material, otherwise it's a manufacture issue, not scientific issue.

4. This statement "The standard reference film should be suitable to the calibration and self-calibration for all instruments" is not correct. And reference 12 is not related to topic.
5. Please check all the references, and make sure they are relevant.

Reviewer: 2

Comments to the Author(s)

In this work, a reference film was studied to improve the oxygen gas transmission measurement accuracy of plastic materials for pharmaceutical packaging. The manometric method and coulometric method was first used to test oxygen gas transmission rate of candidate reference film simultaneously. Besides, the homogeneity, stability, jointly determined value and uncertainty evaluation of the reference film were also discussed. However, some revisions should be made to make the paper more convincing.

1. Check the writing format carefully to ensure its correctness. At the 35 lines on the first page and 23 lines on the second page, what is the "(.1)" and "(.2)" mean? You might want to write "(Figure 1)" and "(Figure 2)". At the the 43 lines on the second page, " $1.01 \times 105\text{Pa}$ " need to be changed to " $1.01 \times 105\text{Pa}$ ". In addition, you had better to indent the first line and align the two ends of article.
2. As for the candidate reference film, how did you prepare your material? The preparation is just cut the film that you bought? If the material is not your original, you can't write "A novel reference film was developed".
3. Please describe your experiment in detail to make sure that others can repeat your experiment. You could name your prepared reference film to distinguish between other materials.
4. In the figure 1, please write the meaning and unit of horizontal and vertical coordinates.
5. Please show all the experiment data using table or graph, includ stability experiment and homogeneity experiment.

Author's Response to Decision Letter for (RSOS-180806.R0)

See Appendix A.

RSOS-190142.R0

Review form: Reviewer 1

Is the manuscript scientifically sound in its present form?

Yes

Are the interpretations and conclusions justified by the results?

Yes

Is the language acceptable?

Yes

Is it clear how to access all supporting data?

Yes

Do you have any ethical concerns with this paper?

No

Have you any concerns about statistical analyses in this paper?

No

Recommendation?

Major revision is needed (please make suggestions in comments)

Comments to the Author(s)

My apology for not fully understanding the issue, it is a little interesting that ASTM itself would say the interlaboratory variations is poor for its published standard method. It is useful and meaningful to characterize a universal reference standard for oxygen gas transmission measurements. However, there is still a couple of issues need to be resolved.

1. It is not clear where material CCRM195017 comes from in the latest version of manuscript, was it the polyester film from Toray Industries?

2. It is a little puzzling that initially the authors introduced the problem of big interlaboratory variations for oxygen gas transmission rate measurement in both the standard and data they collected from 22 Chinese laboratories. However, when CCRM195017 was sent for testing, multiple laboratories using both manometric and coulometric methods gave very consistent results. What happened/changed or am I missing something here. If multiple laboratories can provide consistent results for the same material, it means whatever reference they are using is working properly.

3. What is the advantage of CCRM195017 compare to reference films from Mocon.

Review form: Reviewer 2

Is the manuscript scientifically sound in its present form?

Yes

Are the interpretations and conclusions justified by the results?

Yes

Is the language acceptable?

Yes

Is it clear how to access all supporting data?

Yes

Do you have any ethical concerns with this paper?

No

Have you any concerns about statistical analyses in this paper?

No

Recommendation?

Accept as is

Comments to the Author(s)

In this revised manuscript, the introduction and citation of the proposed new standard reference film for oxygen gas transmission measurement has been completed and modified. And more explanation of the stability experiment and homogeneity experiments has been provided. In addition, more detailed information of the prepared materials has been presented, making the paper more substantial and convincing. The proposed work is suggested to be accepted for publication in this journal.

Decision letter (RSOS-190142.R0)

08-Feb-2019

Dear Dr xie:

Title: A new standard reference film for oxygen gas transmission measurements
Manuscript ID: RSOS-190142

The editor assigned to your paper has now received comments from reviewers. We would like you to revise your paper in accordance with the referee and Subject Editor suggestions which can be found below (not including confidential reports to the Editor). Please note this decision does not guarantee eventual acceptance.

Please submit a copy of your revised paper before 03-Mar-2019. Please note that the revision deadline will expire at 00.00am on this date. If we do not hear from you within this time then it will be assumed that the paper has been withdrawn. In exceptional circumstances, extensions may be possible if agreed with the Editorial Office in advance. We do not allow multiple rounds of revision so we urge you to make every effort to fully address all of the comments at this stage. If deemed necessary by the Editors, your manuscript will be sent back to one or more of the original reviewers for assessment. If the original reviewers are not available we may invite new reviewers.

RSC Associate Editor
Comments to the Author:
(There are no comments.)

Reviewers' Comments to Author:
Reviewer: 2

Comments to the Author(s)

In this revised manuscript, the introduction and citation of the proposed new standard reference film for oxygen gas transmission measurement has been completed and modified. And more explanation of the stability experiment and homogeneity experiments has been provided. In addition, more detailed information of the prepared materials has been presented, making the paper more substantial and convincing. The proposed work is suggested to be accepted for publication in this journal.

Reviewer: 1

Comments to the Author(s)

My apology for not fully understanding the issue, it is a little interesting that ASTM itself would say the interlaboratory variations is poor for its published standard method. It is useful and meaningful to characterize a universal reference standard for oxygen gas transmission measurements. However, there is still a couple of issues need to be resolved.

1. It is not clear where material CCRM195017 comes from in the latest version of manuscript, was it the polyester film from Toray Industries?
2. It is a little puzzling that initially the authors introduced the problem of big interlaboratory variations for oxygen gas transmission rate measurement in both the standard and data they collected from 22 Chinese laboratories. However, when CCRM195017 was sent for testing, multiple laboratories using both manometric and coulometric methods gave very consistent results. What happened/changed or am I missing something here. If multiple laboratories can

provide consistent results for the same material, it means whatever reference they are using is working properly.

3. What is the advantage of CCRM195017 compare to reference films from Mocon.

Author's Response to Decision Letter for (RSOS-190142.R0)

See Appendix B.

Decision letter (RSOS-190142.R1)

25-Feb-2019

Dear Dr Xie:

Title: A new standard reference film for oxygen gas transmission measurements

Manuscript ID: RSOS-190142.R1

It is a pleasure to accept your manuscript in its current form for publication in Royal Society Open Science. The chemistry content of Royal Society Open Science is published in collaboration with the Royal Society of Chemistry.

RSC Associate Editor
Comments to the Author:
(There are no comments.)

Reviewer(s)' Comments to Author:

Appendix A

REVIEWER(S) REPORTS:

Associate Editor Comments to Author ():

RSC Associate Editor:

Comments to the Author:

(There are no comments.)

RSC Subject Editor:

Comments to the Author:

(There are no comments.)

Reviewers' Comments to Author:

Reviewer: 1

Comments to the Author(s)

The authors developed a new standard reference film for oxygen gas transmission measurement. The data showed the material itself is indeed stable and can be used as reference film, however, the problem the authors think they are going to solve may not exist. The fact that they sent their reference film to 4 CNAS laboratories and got consistent results proved the existing reference materials were working properly.

Reply: We appreciated the reviewer's comment. However, with due respect, we disagree. As stated in our manuscript, the calibration membranes for oxygen transmittance in the Chinese market are supplied by instrument manufacturers. These reference membranes can be divided into two categories, calibration membranes for manometric method instrument and those for coulometric method instrument. Each membrane is either for the specific manometric method instrument or the specific calibrating coulometric method instrument only, which causes the different calibration

values and measurement results. This can be evidenced with the inter-laboratory data comparison among 22 Chinese laboratories organized by CCFDA institute. In this inter-laboratory test, the oxygen permeability of the same homogeneous plastic film was measured by either manometric or coulometric method in 22 laboratories. The data obtained by two methods with different instrument exhibited significant variations and dispersion (Table 1 and figure 1) due to the inconsistent traceability reference materials.

Therefore, we discussed this instrument traceability problem with the collaborative laboratories many times, and reached a consensus that the instruments based on the manometric method and the coulometric method should be calibrated with the same reference material with the same quantity value. After thorough research, we selected reference material the calibration membrane for coulometric instruments produced by Mocon. The data obtained with the instruments of either manometric method or coulometric method calibrated with the same reference membrane in the cooperative calibration laboratories exhibited high uniformity and normal distribution. Therefore, in the present work, we developed a general reference membrane, instead of popularizing the calibration membrane for coulometric instrument as reference membrane because: 1. The calibration film is for the coulometric instruments only because of its specific size that is not universal for all instruments; 2. The uncertainty of the universal reference membrane developed in our work includes the uncertainties of both the coulometric method and the manometric method.

1. Reference 7 talked about a novel PLA/PHBV Films and its characterization (including measuring helium permeability, but not oxygen). It is a little puzzling that the authors used this to draw a conclusion that there is dispersion in measuring oxygen transmission rates by ASTM.

Reply: Thank you for pointing this wrong citation. We have added the closely related reference to support our statement. ASTM points out in a regulation item that the data dispersion of oxygen transmittance is large.

2. Data source for table 1 is not clear? Did the authors send the same plastic film to various labs for testing or obtain data from these labs for the same type of material? If it is the later, maybe they are measuring slightly different materials.

Reply: Thank you for your question. The same film samples were sent to different laboratories after the film passed homogeneity and short-term stability tests. The homogeneity and the short-term stability tests have excluded the data variations caused by the unevenness of the sample and transport conditions. Therefore, the dispersion of data variation obtained in the 22 laboratories (Table 1) is much wider than that caused by the unevenness of samples and the transport conditions. We also collected information of the type of instrument used and the traceability calibration membrane used in each laboratory, and found that the dispersion of data collected in these laboratories was attributed to the different traceability of instrument. Based statistic results and the experimental results obtained in our lab with the oxygen transmittance meters of most popular brands based on different methods, we conclude that the significantly discrete data of the 22 laboratories is due to the different traceability of instrument.

3. Reference 4 didn' t mention anything about results from manometric method are lower than coulometric method. The fact that these two methods used different mechanism is exactly the reason why they used calibration films.

Based on 1-3, the problem the authors try to solve may not exist. In fact, any measurement instruments should ship with consistent and known value reference/calibration material, otherwise it' s a manufacture issue, not scientific issue.

Reply: We appreciate the reviewer's comment. I agree that ref. 4 did not give the exact conclusion that the results of coulometric method are higher than those obtained with the of manometric method. However, in the reference 4, the data obtained with the

manometric method are higher than those obtained with the coulometric method. We also found the same pattern in our daily tests. Technically, the principles and conditions of the coulometric method and the manometric method are same, and thus reference materials with the same values should be used. However, different reference materials are provided by the manufacturers of different instruments based on the measurement methods. There is a gap between the values of calibration materials of the two methods. We found that the oxygen transmittance measured by the manometric method is usually lower than that obtained with the coulometric method, possibly because the oxygen concentration of the test side is decreased in the late stage of the manometric measurement. Therefore, it is necessary to unify the traceability of two methods, which is not only a scientific problem, but also the management problem of the unified measurement data. Discussing and understanding the measurement principle, evaluating the consistency of the testing conditions, reliability, and traceability calibration membrane selection are scientific issues. The administrative management is mainly focused on the rationality and unification of the test data, the instrument design, quality control and quality evaluation of the products.

4. This statement “The standard reference film should be suitable to the calibration and self-calibration for all instruments” is not correct. And reference 12 is not related to topic.

Reply: Thank you for careful reading. The sentence has been revised in the revised manuscript. Instrument calibration is used to correct the measurement data. Self-calibration is used to check the reliability of instrument and operation. We meant to state that we have developed a universal reference membrane applicable to all oxygen transmittance gauges, not for a specific method or instrument. We have checked reference 12 and corrected it.

5. Please check all the references, and make sure they are relevant.

Reply: We have checked all the references to make sure they are relevant to our work, as suggested by the reviewer.

Reviewer: 2

Comments to the Author(s)

In this work, a reference film was studied to improve the oxygen gas transmission measurement accuracy of plastic materials for pharmaceutical packaging. The manometric method and coulometric method was first used to test oxygen gas transmission rate of candidate reference film simultaneously. Besides, the homogeneity, stability, jointly determined value and uncertainty evaluation of the reference film were also discussed. However, some revisions should be made to make the paper more convincing.

1. Check the writing format carefully to ensure its correctness. At the 35 lines on the first page and 23 lines on the second page, what is the “.1)” and “.2)” mean? You might want to write “.1)” and “.2)”. At the the 43 lines on the second page, “ $1.01 \times 105\text{Pa}$ ” need to be changed to “ $1.01 \times 10^5\text{Pa}$ ”. In addition, you had better to indent the first line and align the two ends of article.

Reply: Thank you for your suggestion. We have read the whole manuscript carefully and corrected the writing format. The first line and the two ends of the manuscript have been aligned.

2. As for the candidate reference film, how did you prepare your material? The preparation is just cut the film that you bought? If the material is not your original, you can't write “A novel reference film was developed” .

Reply: Thank you for your questions. For developing such candidate reference film, we did more than just cutting the film. First, we screened the films various plastic films on the market with a number of quality indexes including the amount, homogeneity,

and stability of oxygen transmissivity, and the homogeneity of thickness. Once the candidate films chosen, we optimize the size of film based on the sampling requirements of universal instruments. The candidate film was then measured and calibrated based on the ISO guidelines. However, to avoid any misunderstanding, we have revised the sentence to "A novel reference film was characterized".

3. Please describe your experiment in detail to make sure that others can repeat your experiment. You could name your prepared reference film to distinguish between other materials.

Reply: Thank you for your suggestions. We have described the detailed experimental procedures in the revised manuscript. The candidate reference film is given the name CCRM195017.

4. In the figure 1, please write the meaning and unit of horizontal and vertical coordinates.

Reply: Thank you for your careful reading. We have supplemented the unit and meaning of the horizontal and vertical ordinates in Figure 1 in the revised manuscript.

5. Please show all the experiment data using table or graph, include stability experiment and homogeneity experiment.

Reply: Thank you for your comments. We have included the homogeneity and stability of data in revised manuscript.

Appendix B

Reviewers' Comments to Author:

Reviewer: 2

Comments to the Author(s)

In this revised manuscript, the introduction and citation of the proposed new standard reference film for oxygen gas transmission measurement has been completed and modified. And more explanation of the stability experiment and homogeneity experiments has been provided. In addition, more detailed information of the prepared materials has been presented, making the paper more substantial and convincing. The proposed work is suggested to be accepted for publication in this journal.

Reply: We appreciate your comments and thank you for the recommendation for publication.

Reviewer: 1

Comments to the Author(s)

My apology for not fully understanding the issue, it is a little interesting that ASTM itself would say the interlaboratory variations is poor for its published standard method. It is useful and meaningful to characterize a universal reference standard for oxygen gas transmission measurements. However, there is still a couple of issues need to be resolved.

1. It is not clear where material CCRM195017 comes from in the latest version of manuscript, was it the polyester film from Toray Industries?

Reply: Thank you for the question. Material CCRM195017 is the polyester film purchased from Toray Industries. Our company can characterize reference materials but cannot produce membrane materials. We collected a variety of commercial membrane materials and screened them for the uniformity. The polyester film produced by Toray Industries exhibited the best homogeneity for oxygen permeation. Therefore, it was further characterized as the reference membrane.

2. It is a little puzzling that initially the authors introduced the problem of big interlaboratory variations for oxygen gas transmission rate measurement in both the standard and data they collected from 22 Chinese laboratories. However, when CCRM195017 was sent for testing, multiple laboratories using both manometric and coulometric methods gave very consistent results. What happened/changed or am I missing something here. If multiple laboratories can provide consistent results for the same material, it means whatever reference they are using is working properly.

Reply: This is a good question. As stated in our manuscript, based on the extensive investigation and discussion with other labs, we concluded that the big interlaboratory variations for oxygen gas transmission rate measurement in the data we collected from 22

Chinese laboratories were due to the inconsistent traceability reference materials. We discussed this instrument traceability problem with the collaborative laboratories and reached a consensus that both instruments based on the manometric method and the coulometric method should be calibrated with the same reference material with the same quantity value. After thorough investigation, we selected reference membrane for coulometric instruments produced by Mocon as the universal reference membrane. These laboratories gave very consistent results with either manometric or coulometric method after adopting the reference membrane we proposed.

3. What is the advantage of CCRM195017 compare to reference films from Mocon.

Reply: Thank you for your question. As stated in our manuscript, the CCRM195017 and reference films produced by Mocon possess similar oxygen transmission quantity value, but slightly different uncertainty and significantly different sizes. The differences in uncertainty and size make CCRM195017 more universal.

1. The uncertainty of CCRM195017 (relative uncertainty is 6.6%) includes the uncertainty introduced by the manometric method and the coulometric method. The reference films from Mocon included only the uncertainty introduced by the coulometric method (relative uncertainty is 5%). Therefore, CCRM195017 is suitable for calibrating both instruments of the manometric method and coulometric method. In contrast, the reference films from Mocon are more suitable for the calibration of the instrument based on coulometric method.

2. CCRM195017 reference membrane is designed with a large square shape of 15cm*15cm, which can be cut into different sizes based on the different gas transmitters. The reference films from Mocon are designed for the Mocon oxygen transmittance meter only. Their size cannot be changed, and thus they are more suitable for instruments of Mocon Company or the instruments with similar test chambers to those of the Mocon instruments.